

# Observations of anomalous propagation over waters near Sweden

Lars Norin[1]

[1]Radar Electronic Warfare Systems, Department for C4ISR, Swedish Defence Research Agency, Linköping, Sweden

**Correspondence:** Lars Norin (lars.norin@foi.se)

**Abstract.** Radio waves propagating in the atmosphere are affected by the prevailing atmospheric state. The state of the atmosphere can cause radio waves to refract more or less towards the ground. When the refractive index of the atmosphere differs from standard atmospheric conditions the propagation is called anomalous. Radars which are affected by anomalous propagation can receive ground clutter far beyond the radar horizon. In this work 4.5 years of data from five operational Swedish C-band dual polarization weather radars are presented. Analyses of the data reveal a strong seasonal cycle and weaker diurnal cycle in ground clutter from coastal regions across nearby waters. The impact of anomalous propagation on ground clutter, measured with horizontal and vertical polarization, was compared but no clear difference was found.

## 1 Introduction

The atmosphere has a large impact on the propagation of electromagnetic waves. Radio waves are, for example, attenuated by precipitation but due to the atmosphere's refractive property radio waves can also be refracted. The refractive index of the atmosphere depends on the temperature, pressure, and water vapor (Bean and Dutton, 1966; Battan, 1973; ITU, 2019). Due to the vertical inhomogeneity of the atmosphere radio waves are normally refracted slightly towards the ground. One typical such atmospheric state is referred to as the "standard atmosphere" (see, e.g., Patterson, 2008). However, other atmospheric conditions can also occur during which radio waves are refracted less (subrefraction) or more (superrefraction) towards the ground. In some cases radio waves can even become trapped and reach the ground far beyond the normal radio horizon. These "non-standard" atmospheric conditions lead to what is known as anomalous propagation (Battan, 1973; Turton et al., 1988).

Effects of anomalous propagation of radio waves was known already in the 1930s (Kerr, 1951). Since then anomalous propagation has been studied thoroughly (see, e.g., Kerr, 1951; Bean and Dutton, 1966; Battan, 1973) and the research field is still active. For example, in order to better understand how the atmosphere's refractive index changes with height a number of studies based on in situ measurements have been conducted. The refractive index has been measured at different heights using radiosondes (e.g. Steiner and Smith, 2002; Bech et al., 2002; Wang et al., 2018), or less commonly, using instruments attached to towers or masts (Falodun and Ajewole, 2006; Adediji et al., 2011) or helicopters (Babin, 1996).

In situ measurements can achieve very high vertical resolution and in some cases (e.g. measurements from towers or masts) high temporal resolution over long time periods. However, the horizontal resolution has so far been limited for in situ measurements. With the advance of numerical weather prediction models (NWP) a new way of investigating the atmosphere's refractive index has become available. von Engeln and Teixeira (2004) and Lopez (2009) used data from the European Centre





for Mid-Range Weather Forecasting (ECMWF) to present global climatologies of the atmosphere's refractive index. Many more studies have used NWP data to examine the refractive index in more limited geographical regions (Atkinson and Zhu, 2006; Sirkova, 2015; Magaldi et al., 2016; Emmanuel et al., 2017). NWP data have been compared with radiosonde data and shown good agreement (von Engeln and Teixeira, 2004; Bech et al., 2007; Zhu et al., 2022).

Anomalous propagation can have a large impact on radar systems. If a radar beam is subjected to atmospheric subrefraction, targets close to the ground can be missed whereas if a radar beam is superrefracted, unexpected ground clutter can occur far beyond the normal radar horizon. In addition to changes in distances at which a target can be detected and increases in ground clutter, a range-height error can also occur (Skura, 1987). A range-height error can lead to an erroneous estimation of a target's true height. For the radar user it is therefore important to know when and how often anomalous propagation conditions occur.

For weather radars anomalous propagation often leads to unwanted ground clutter (Doviak and Zrnić, 2006). To address this problem much work has been done to derive various algorithms that can detect and remove such echoes (see, e.g., Moszkowicz et al., 1994; Grecu and Krajewski, 2000; Cho et al., 2006; Overeem et al., 2020; Husnoo et al., 2021). Even though most studies concerning weather radars and anomalous propagation have focused on mitigating the effects of ground clutter, data from weather radars have also been used to study anomalous propagation itself. Fornasiero et al. (2006) analyzed three years of data from two C-band dual polarization weather radars in northern Italy and found a seasonal cycle as well as a diurnal cycle in the received ground clutter. Increased occurrence of superrefraction was observed in the summer as well as during nights and mornings. Mesnard and Sauvageot (2010) analyzed one year of data from an S-band weather radar in southwest France. They reported on the spatial distribution and the temporal duration of continuous ground clutter detected far beyond the radar horizon from both sea and land.

Anomalous propagation is common in littoral environments (see, e.g., von Engeln and Teixeira, 2004). Even though Sweden is surrounded by waters (e.g. the Baltic Sea, the Gulf of Bothnia, and Kattegat) only a few studies on anomalous propagation based on data from this region have been published. Alberoni et al. (2001) proposed a method for removing ground clutter and sea clutter from weather radars and presented results from applying this technique on a few days worth of data from a weather radar on the Swedish island Gotland in the Baltic Sea. A number of measurement campaigns have also been conducted in the Baltic sea, mainly focused on studying evaporation ducts. During these campaigns a combination of in situ measurements, radiosondes, and microwave measurements were performed and reported (see, e.g., Scholz and Förster, 2003; Förster et al., 2004; Essen et al., 2004; Förster and Riechen, 2006; Essen et al., 2012; Danklmayer et al., 2013, 2015, 2016a, b).

    Despite the large impact of anomalous atmospheric conditions on the propagation of radio waves, no long term analyses of continuous radar measurements over the Baltic Sea or its neighboring waters have, as far as the author is aware, been published. A more thorough investigation of anomalous propagation in this region is therefore warranted. In this work we present data from five Swedish weather radars from 2017 to 2021 in order to study the seasonal and diurnal cycles of anomalous propagation over the waters near Sweden.

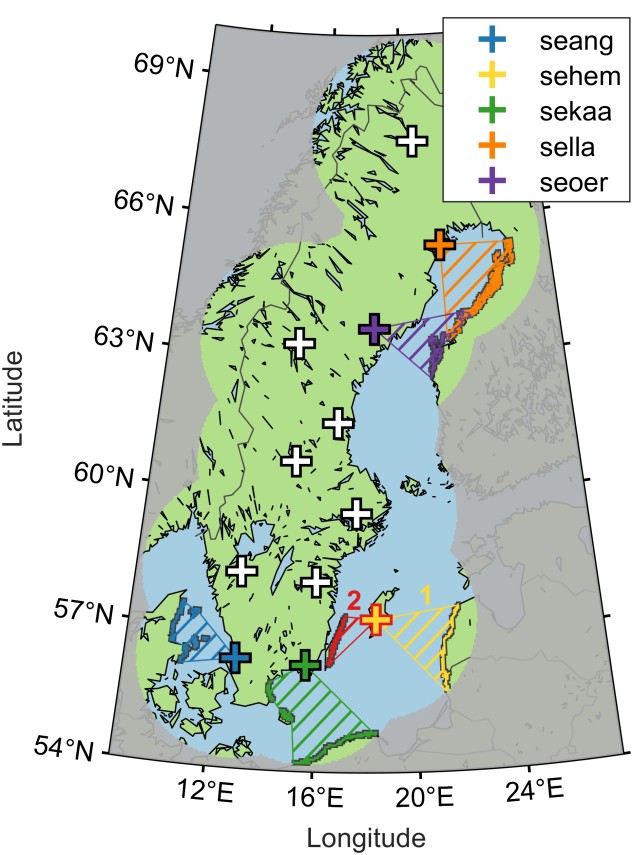

**Figure 1.** The Swedish weather radar network and its coverage. Radar locations are marked by plus signs. The five radars used in this work are listed in the legend. Six groups of radar cells from these five radars were selected to monitor for ground clutter. These radar cells are shown in different colors.

## 2   Data

The Swedish weather radar network consists of 12 dual polarization, C-band, Doppler weather radars. The radars provide almost nationwide cover with an update rate of 5 min. The current weather radars were recently modernized from single to dual polarization. The modernization dates for the radars that were used in this work are shown in Table 1.

Weather radars measure echo strength using the equivalent radar reflectivity factor Z (hereafter called reflectivity) which is expressed in $\mathrm{mm}^6\mathrm{m}^{-3}$ or, in its corresponding logarithmic unit, dBZ (Doviak and Zrnić, 2006). The minimum echo strength

the Swedish weather radars register is $-32$ dBZ. Any signal equal to or below the minimum echo strength is given this value. This echo value is later in this work referred to as "no echo". The maximum echo strength the radars record is 96 dBZ.

The scan strategy consists of 10 different elevation angles, with $0.5°$ being the lowest elevation angle. The maximum unambiguous range for the lowest four elevation angles is 240 km. The radar data are stored with a range resolution of 250 m and



**Table 1.** Location and modernization dates for the weather radars selected for this work.

| Radar name | Node ID | Date of modernization | Latitude (° N) | Longitude (° E) | Altitude (m) |
|---|---|---|---|---|---|
| Ängelholm | seang | 21 Jan 2017 | 56.3675 | 12.8517 | 207 |
| Hemse | sehem | 27 Jun 2017 | 57.3034 | 18.4003 | 85 |
| Karlskrona | sekaa | 20 Dec 2018 | 56.2955 | 15.6102 | 130 |
| Luleå | sella | 20 Sep 2018 | 65.4309 | 21.8650 | 47 |
| Örnsköldsvik | seoer | 2 Oct 2017 | 63.6395 | 18.4019 | 508 |

**Table 2.** Some of the technical parameters used by the Swedish weather radar systems.

| Parameter | Value |
|---|---|
| Transmit power | 250 kW |
| Gain | 45 dB |
| Beamwidth | $1.0°$ |
| Frequency | 5.6 GHz |
| Rotational speed | $18° \, \mathrm{s}^{-1}$ |
| Maximum range | 240 km |
| Range resolution | 250 m |
| Azimuthal resolution | $1.0°$ |
| Lowest elevation angle | $0.5°$ |
| Update time | 5 min |

have an azimuthal resolution of $1°$. These and some other technical parameters of the radars are shown in Table 2. For more

information on the radars, see the World Meteorological Organization weather radar database[1].

According to the radar program of the European Meteorological Services Network (EUMETNET OPERA[2]) weather radars should be identified by a five letter "node" (Michelson et al., 2019). The node identifiers for the Swedish radars are shown in Table 1 as well as in Fig. 1. The node identifiers are used to refer to the different radars throughout the rest of this document.

The Swedish weather radars generate a range of products based on the received echoes. Two products have been used in this

work: the total (unfiltered) reflectivity data and the corresponding Doppler filtered reflectivity data. Since the radars use dual polarization the impact of anomalous propagation on polarization has also been studied. For this reason data sets from both horizontal and vertical polarization were used. The weather radars use dual polarization in a simultaneous transmit and receive mode. Only data from the lowest elevation angle, $0.5°$, were used. Data from 1 January 2017 until 20 July 2021 were analyzed.

---

[1]https://wrd.mgm.gov.tr/Home/Wrd
[2]https://www.eumetnet.eu/activities/observations-programme/current-activities/opera/



## 3 Method

### 3.1 Extracting ground clutter from weather radar data

As was mentioned in Sect. 2 the Swedish weather radars generate two products that are of interest for this study: the total reflectivity data and the Doppler filtered reflectivity data. The Doppler filtered data are useful for measuring precipitation rates, as this filter suppresses echoes from non-moving targets such as ground clutter. The Doppler filtered data is the product that is of most interest for meteorological analyses. The total reflectivity data are useful for comparison and for troubleshooting but are otherwise normally of little interest for meteorological purposes.

In this work we are to the contrary only interested in ground clutter and wish to suppress echoes originating from other sources. By subtracting the Doppler filtered reflectivity data from the total reflectivity data, ground echoes can be extracted. In addition many spurious signals, such as emissions from the sun or from man-made transmitters, can also be suppressed as long as these signals deviate sufficiently from the radar's own frequency.

Figure 2 shows reflectivity data from weather radar *sehem*, located on the island Gotland in the Baltic Sea. The data presented in the figure have been averaged over the summer months (June, July, and August) from 2017 to 2021. Different panels in Fig. 2 show the total reflectivity data, the Doppler filtered reflectivity data, and the difference between these two data sets.

Figure 2a shows the total reflectivity data. Here the most prominent echoes originate from cargo ships, fishing vessels, etc. that regularly traffic the Baltic Sea. The ship routes are clearly seen, extending from the southwest to the northeast of the island. Strong echoes also come from ground clutter near the coastline of the Swedish island Öland to the southwest and from the coastlines of Latvia and Lithuania to the east. Interference from other transmitters can also be seen, showing up as spokes from the edges to the radar at the center. In the background a much weaker but more homogeneous echo can be seen. This background echo is the result of precipitation.

Figure 2b shows the Doppler filtered reflectivity data. In this data set much of the ground clutter from the coastal regions have been suppressed but most of the ship echoes and interference from other transmitters still remain. However, when the radial velocities of the ships are close to zero most of these echoes are suppressed. The precipitation background echoes remain in this data set.

Figure 2c shows the difference between total and filtered reflectivity data. From this data set it is seen that the ground clutter from the coastlines are back. The echoes from the ships are suppressed (except when the radial velocities of the ships are close to zero). The background echoes from precipitation have also been reduced while interference spokes from other transmitters to a large degree still remain. This data set (the difference between the total and the Doppler filtered reflectivity data) is used for further analyses.

Figure 2d shows two groups of radar cells that were selected for radar *sehem* for further study (see Sect. 3.2). The radar horizon, calculated using standard atmospheric conditions, is also shown for comparison. It can be seen that all the selected radar cells are located well beyond the radar horizon.

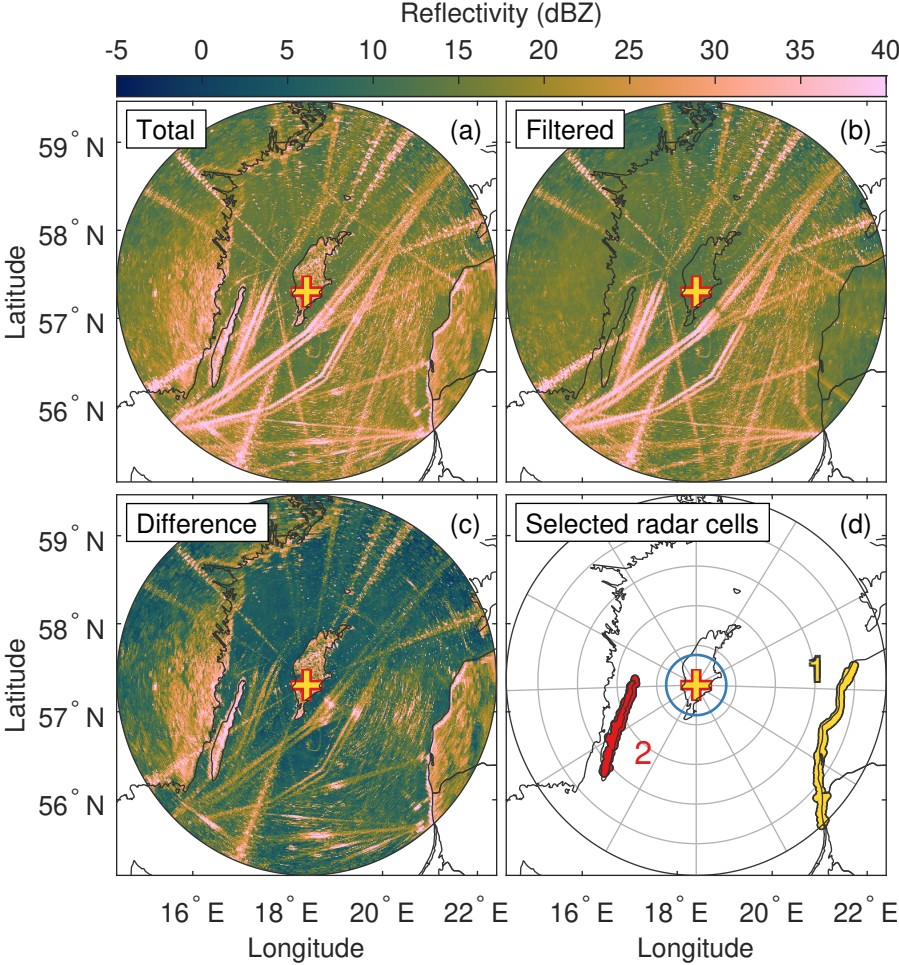

**Figure 2.** Data from weather radar *sehem* on the island Gotland in the Baltic Sea, averaged over June, July, and August 2017–2021. Top left (a): Total reflectivity data. Top right (b): Doppler filtered reflectivity data. Bottom left (c): Difference between total and filtered reflectivity data. Bottom right (d): Radar cells from two regions (the island Öland to the southwest and the coastlines of Latvia and Lithuania to the east) were selected to study the effects of anomalous propagation. The blue ring near the radar shows the radar horizon, assuming standard atmospheric conditions. Range rings are displayed at 50 km increments.

## 3.2 Selecting radar cells to monitor for ground clutter

The radars that are most interesting for this study are radars that are situated near Sweden's coastline and for which land exist, within their maximum range, on the far side of nearby waters. Five radars were found to fulfill these conditions. These radars are shown in Fig. 1 and are also listed in Table 1.

For each of the found radars, cells from the coastline of the far side of nearby waters were selected. The selected radar cells all showed large reflectivity values compared to neighboring radar cells, when averaged over summer months (June to August,



**Table 3.** Number of radar cells in the selected regions together with minimum, median, and maximum distances to these cells.

| Group of radar cells | Number of radar cells | Distance (km) | | |
|---|---|---|---|---|
| | | Min. | Median | Max. |
| seang | 796 | 84 | 131 | 188 |
| sehem 1 | 379 | 170 | 188 | 240 |
| sehem 2 | 408 | 76 | 106 | 162 |
| sekaa | 672 | 95 | 135 | 240 |
| sella | 2374 | 133 | 165 | 231 |
| seoer | 596 | 136 | 165 | 236 |

cf. Fig. 2). The selected radar cells are shown in Figure 1. For the radar on the island Gotland, *sehem*, radar cells were selected both to the east (the coastlines of Latvia and Lithuania) and to the west (the island Öland).

Table 3 lists the number of radar cells that were selected for each radar together with the minimum, median, and maximum distances to these cells.

### 3.3 Data analyses

In order to analyze the effects of anomalous propagation, the difference data set (see Sect. 3.1) was used. For all data analyses the difference data set with horizontal polarization from the lowest elevation angle was used, unless otherwise stated.

From the difference data set reflectivity values from the selected radar cells were extracted. For each group of radar cells time series and histograms of the reflectivity data were generated. The time series were constructed by calculating the median reflectivity value from all radar cells in each group for every time step (i.e. with five minute time resolution). In order to analyze the distribution of the reflectivity values, four different types of histogram were generated.

In the first type of histogram the values of the reflectivity data from all the groups of radar cells were stored for each day of the year. Reflectivity data were binned with $0.5\,\mathrm{dBZ}$ resolution, i.e. stored in 256 bins. The histograms thus resulted in a matrix for each selected group with the size $365 \times 256$. From the histograms empirical cumulative distribution functions (ECDFs) of reflectivity data were calculated for each day. In addition, a time series of binary values was constructed, representing "echo" (reflectivity values $> -32\,\mathrm{dBZ}$) or "no echo" (reflectivity values $= -32\,\mathrm{dBZ}$) .

In the second type of histogram the values of all reflectivity data from all the groups of radar cells were stored as a function of the time of day (resulting in matrices with the size $288 \times 256$, i.e. the daily time resolution times the resolution of the binned data). In the same way as for the previous histograms ECDFs were calculated for each time step of the day. Also, time series of binary values were constructed, representing "echo" or "no echo".

In the third type of histogram the relative frequency of echoes (i.e. reflectivity values $> -32\,\mathrm{dBZ}$) for all groups of radar cells were calculated as a function of day of the year and as a function of the time of the day (resulting in matrices with the size $365 \times 288$, i.e. the number of days of the year times the daily time resolution).





The fourth type of histogram consisted of pairwise reflectivity values from each of the three data sets (i.e. total, Doppler filtered, and difference), using horizontal and vertical polarization. For each group of radar cells the number of measurements with a certain horizontal reflectivity value and a certain vertical reflectivity value were stored (resulting in matrices with the size $256 \times 256$, i.e. the resolution of the binned data with horizontal polarization times the resolution of the binned data with vertical polarization).

## 4   Results

### 4.1   Seasonal variation in ground clutter strength

Using the time series that were extracted for each of the selected groups of radar cells, the reflectivity data as a function of time could be visualized. Figure 3 shows the median reflectivity values from the selected groups of radar cells as a function of the time of year together with a line depicting a moving average in time (achieved by applying a 14 day long Hanning weighted window). For many of the selected groups a seasonal cycle can be seen, perhaps most notably for group *sehem 1*. The values of the reflectivity data are for every selected group, on average, higher during the summer months (June, July, August) compared to the winter months (December, January, February). In the figure it can also be seen that the reflectivity data from group *seoer* are much lower than those from the other groups. This is most likely a result of radar *seoer* being situated at a much higher altitude compared to the other radars (cf. Table 1). For radar *seoer* to detect ground clutter from the far side of the Gulf of Bothnia the atmospheric superrefraction must be much stronger than for the other radars.

Another way of examining the time series from the selected groups of radar cells for periodic patterns is to perform a spectral analysis. Spectral analyses of all time series are shown in Fig. 4. The one year period and the one day period are highlighted in the figure. All time series show a peak in the spectral amplitude at the one year period but only a few show signs of a diurnal cycle. Even for the time series in which the diurnal cycle is discernible (e.g. group *sekaa*) it is much weaker compared to the seasonal cycle.

The time series only show median reflectivity values and the seasonal cycle can be examined in greater detail by studying the histograms that were generated in the data analyses (cf. Sect. 3.3). Figure 5 shows the ECDFs of the reflectivity data as a function of the time of year for all selected groups of radar cells. In order to more clearly display the seasonal cycle a moving average in time (achieved by applying a seven day long Hanning weighted window) was applied to the data. From the figure it is clear that an increase in reflectivity values occur during the summer months but that most measurements still belong to the data bin with the lowest reflectivity value (i.e. $-32\,\mathrm{dBZ}$). This value, as explained in Sect. 2, actually shows that no echo of sufficient strength was detected. It can also be seen from Fig. 5 that radar *seoer* detects far fewer echoes from its selected group of radar cells compared to the other radars. For over 90 % of the time there were no echoes detected by radar *seoer* from its selected radar cells.

Instead of examining the ECDFs of the reflectivity data binary time series were generated, representing "echo" or "no echo". Figure 6 shows the relative frequency of detected echoes as a function of the time of year for all selected groups of radar cells. To more clearly show the seasonal cycle a moving average was applied (using a 14 day long Hanning weighted window).



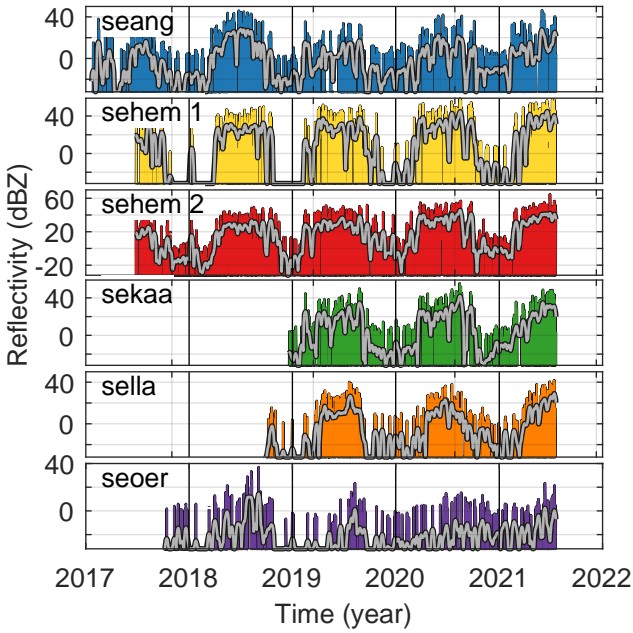

**Figure 3.** Time series of median values of reflectivity data from all selected groups of radar cells. Lines showing a moving average in time are depicted in gray.

From Fig. 6 it is clear that the chance of detecting ground clutter from the selected groups of radar cells increases during the summer months. This is true for all groups but is least clearly visible for group *seoer* due to reason discussed above.

## 4.2 Diurnal variation in ground clutter strength

To investigate the extracted data for the presence of a diurnal cycle the histogram that stored reflectivity values as a function of the time of day can be studied. In the same way as for the seasonal cycle, Fig. 7 shows the ECDFs of reflectivity data as a function of the time of day for all selected groups of radar cells. A weak, but consistent diurnal cycle can be seen for all groups. The ECDFs show a decrease in stronger reflectivity values in the morning (around 6–7 AM, local time) and a slight increase in stronger reflectivity values in the evening (around 18–19 PM, local time). It can also be seen that for a majority of the time no echoes were detected from any group of radar cells. This is particularly evident for echoes from group *seoer* from which detectable echoes were observed less than 5 % of the time.

As for the analysis of the seasonal cycle a binary value can be defined from the data, representing "echo" or "no echo". The relative frequency of detected echoes as a function of the time of day is shown in Fig. 8. Here the diurnal cycle is clearly visible for all groups of radar cells except for group *seoer*. Even for the other groups echoes were only detected on average some 10–25 % of the time and the variation during the day was of the order of 5 % or less.





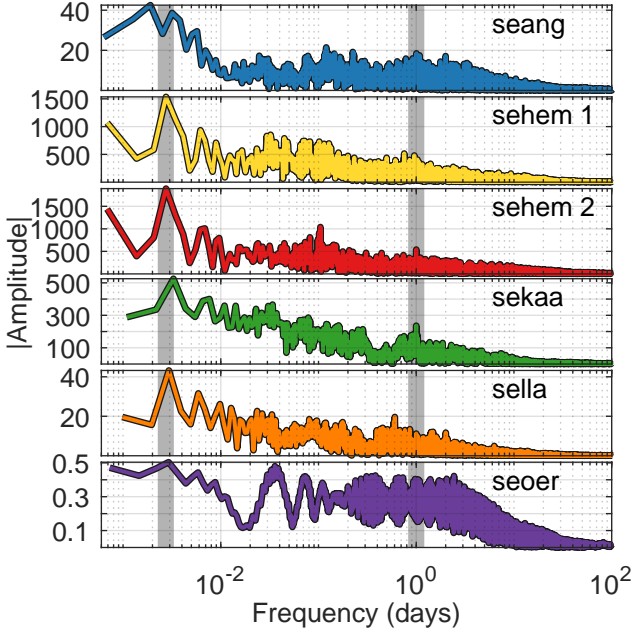

**Figure 4.** Spectral analysis of the median values of the reflectivity data from all radar cells in the selected areas. The one year and one day period are highlighted in the figure with gray vertical lines.

## 4.3 Seasonal and diurnal variation in ground clutter strength

From Fig. 6 and Fig. 8 it is clear that there exist both a seasonal cycle and a diurnal cycle in the reflectivity data, extracted from the selected groups of radar cells. In order to separate the two cycles Fig. 9 shows the relative frequency of detected echoes

both as a function of the time of year as well as a function of the time of day. To improve visibility the data in the figure were smoothed over the time of year by applying a moving average filter (using a seven day long Hanning weighted window). No smoothing was applied to the time of day.

Figure 9 highlights the results that have already been shown. A prominent seasonal cycle is seen for all selected groups of radar cells (even though is it considerably weaker for group *seoer*). Most echoes occurred during summer and echoes were

195 clearly less common during winter. Imposed on the seasonal cycle a diurnal cycle can be seen. Echoes were more frequently detected during the evening, around 18 PM, local time. Fewest echoes were detected in the morning, around 6–7 AM, local time.

## 4.4 Polarization and ground clutter strength

The results presented above were all based on measurements using horizontal polarization. In order to investigate whether

ground echoes from a vertically polarized wave were affected in the same way by anomalous propagation, pairwise measurements of horizontal and vertical polarization were stored.





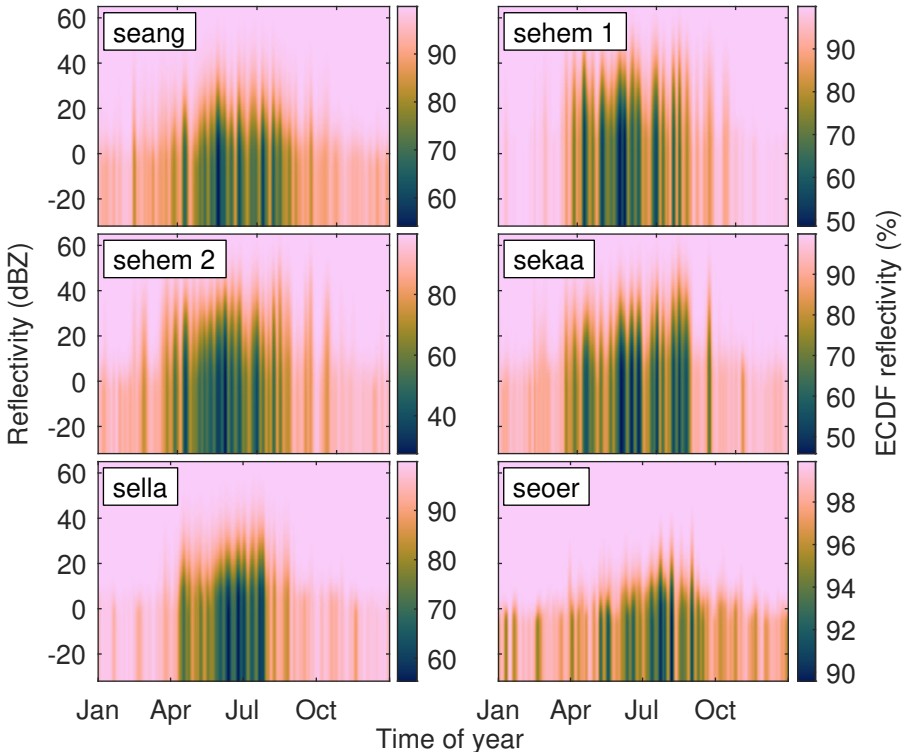

**Figure 5.** Empirical cumulative distribution function of reflectivity values from all selected groups of radar cells as a function of the time of year.

From the pairwise data set two-dimensional histograms were generated for all selected groups of radar cells using the total (unfiltered) data set, the Doppler filtered data set, and the difference data set (see Sect. 3.1). The results are presented in Fig. 10. In the figure the histograms for the three different data sets are shown together with relative frequencies of the

205 difference between the reflectivity values of the two polarizations for all three data sets.

In can be seen from the histograms that most of the data are concentrated along the diagonal line at which the horizontal and the vertical reflectivity values are equal. Some deviations can be seen but the distributions are mostly symmetrical for all three data sets. Data from the Doppler filtered data set is most closely concentrated along the diagonal line, which is true for all groups of radar cells. Since the main difference between the Doppler filtered data set and the other two data sets (at the selected

radar cells) consists of ground clutter, the implication is that ground clutter vary more than echoes from precipitation, when using different polarization. However, from the figure it is difficult to tell if ground clutter from either polarization is stronger, on average.

To find any systematic difference between the ground clutter reflectivity values from the two polarizations, statistical parameters were calculated from the pairwise data set. Table 4 lists the mean ($\mu$), the standard deviation ($\sigma$), and the skewness

($\tilde{\mu} = \mathrm{E}\left[(x-\mu)^3/\sigma^3\right]$) of the pairwise data set for all selected groups of radar cells. The table emphasizes what can be seen





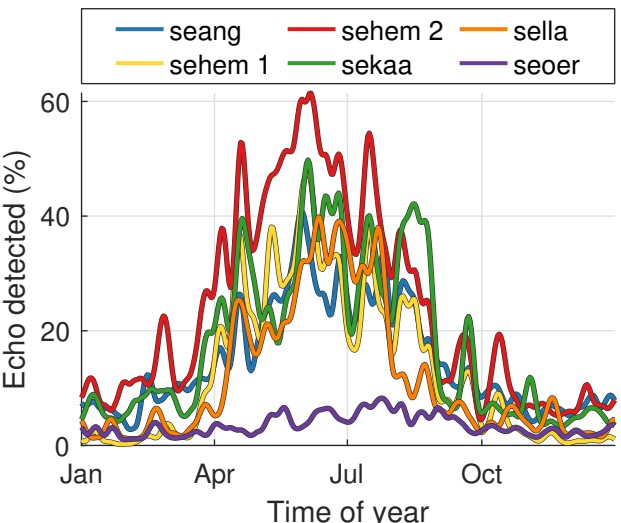

**Figure 6.** Relative frequency of detected echoes from the selected groups of radar cells as a function of the time of year.

**Table 4.** Statistical parameters for the difference in pairwise reflectivity values measured using horizontal and vertical polarization, for all selected groups of radar cells and three different data sets. Mean values ($\mu$) in dB, standard deviation ($\sigma$) in dB, and skewness ($\tilde{\mu}$), unitless.

| Groups of | Total | | | Filtered | | | Difference | | |
|---|---|---|---|---|---|---|---|---|---|
| radar cells | $\mu$ | $\sigma$ | $\tilde{\mu}$ | $\mu$ | $\sigma$ | $\tilde{\mu}$ | $\mu$ | $\sigma$ | $\tilde{\mu}$ |
| seang | 0.29 | 3.88 | -0.08 | 0.44 | 2.23 | 0.27 | 0.21 | 4.38 | -0.03 |
| sehem 1 | 0.97 | 5.36 | 0.17 | -0.05 | 2.03 | 1.19 | 1.03 | 5.49 | 0.13 |
| sehem 2 | -0.02 | 4.47 | 0.02 | -0.10 | 1.54 | 0.68 | 0.02 | 4.66 | -0.01 |
| sekaa | 0.18 | 4.42 | 0.01 | 0.66 | 1.65 | 0.68 | 0.11 | 4.61 | 0.05 |
| sella | 1.13 | 4.08 | 0.55 | 0.21 | 1.47 | 0.38 | 1.18 | 4.36 | 0.39 |
| seoer | -0.17 | 2.52 | 0.58 | -0.09 | 1.75 | 3.30 | -0.15 | 3.29 | 0.22 |

in Fig. 10, namely that the standard deviations for the total and difference data sets are larger than the standard deviation for the Doppler filtered data set. Ground clutter from horizontal polarization tends to result in slightly stronger reflectivity values compared to vertical polarization (up to ca 1 dB). The Doppler filtered data set has the largest skewness values, which are positive for all selected groups of radar cells.





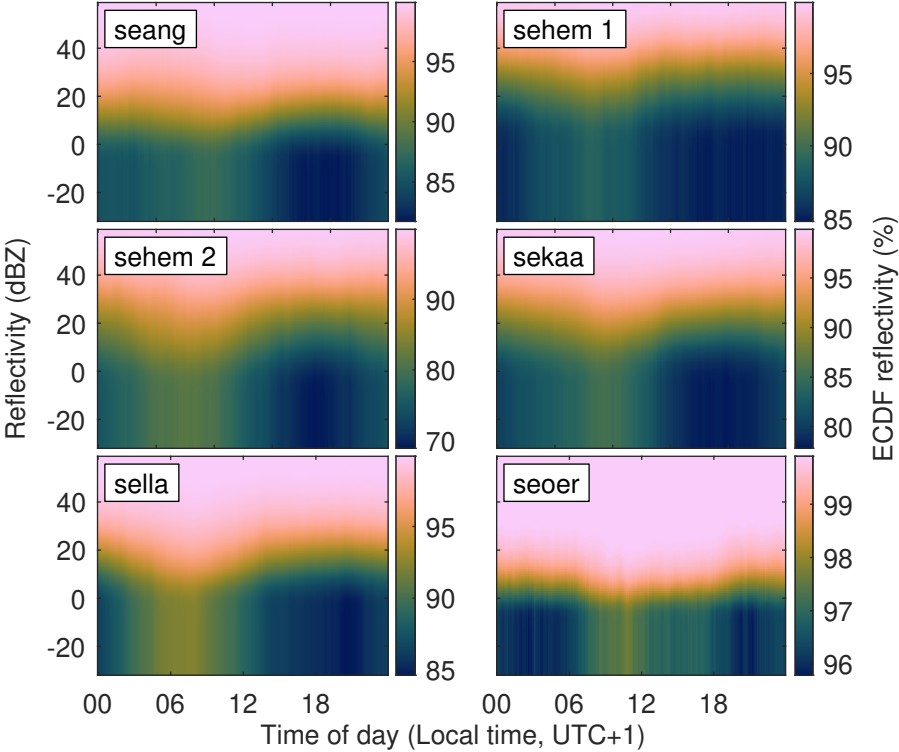

**Figure 7.** Empirical cumulative distribution function of reflectivity values for all selected groups of radar cells as a function of the time of day.

## 5 Discussion

In this work data from operational weather radars have been used to analyze anomalous propagation. To do this echoes from non-moving targets were extracted from radar cells in coastal areas on the far side of nearby waters. While it is not possible to know the radar cross section of the ground in the selected radar cells, the relative changes in echo strength over time can be studied.

The assumption in this work is that the changes in ground clutter strength over time are due to changes in the environment, and in particular to variations in the refractive index of the atmosphere between the radar and the selected radar cells. However, there are other factors that also can change and have an impact on the received echoes.

Precipitation attenuates radar waves and could affect the strength of the received ground clutter. But since precipitation in the studied region generally is higher during the summer and precipitation often is in form of snow in the winter (which leads to weaker reflected echoes), the seasonal cycle should actually be more pronounced if this was taken into consideration.





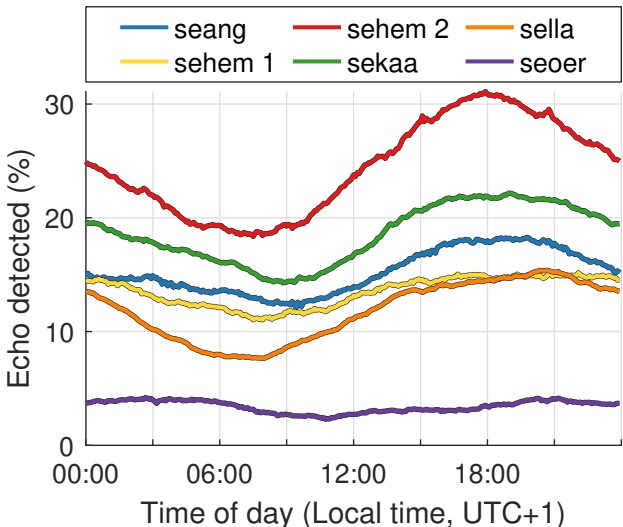

**Figure 8.** Relative frequency of detected echoes from all the selected groups of radar cells as a function of the time of day.

For the period of the study (2017–2021) the Gulf of Bothnia was ice covered for four of the five years. The radar most significantly affected by the ice cover was radar *sella*. A more detailed analysis of how the ice cover affects the atmospheric conditions would be interesting to conduct but is out of scope of this work.

There are some limitations using the presented method. While analyzing the variation in ground clutter over time provides an understanding of how the radars are affected by anomalous propagation, it is not possible to extract the horizontal distribution of the atmospheric refraction. It is quite possible that the refraction is inhomogeneous, the ground clutter only reveal the total impact.

Another limitation using this method is that it is not possible to account for the presence of subrefraction. As the weather radar should not detect ground clutter from far beyond the radar horizon during standard atmospheric conditions there should be no difference in the received ground clutter during subrefractive conditions.

The results presented in this work could be used to compare with atmospheric refractive indices extracted from NWP data, if used together with a radio wave propagation model. The results could also be compared to other data sets from the same region. It would be especially interesting to compare the presented data with a data set based on another frequency such as data from X-band radars or communication signals on the VHF-band.

## 6 Conclusions

In this work 4.5 years worth of data from operational weather radars in Sweden have been used to analyze anomalous propagation. Five radars situated close to the Swedish coast were selected for this study. From these radars, radar cells from six





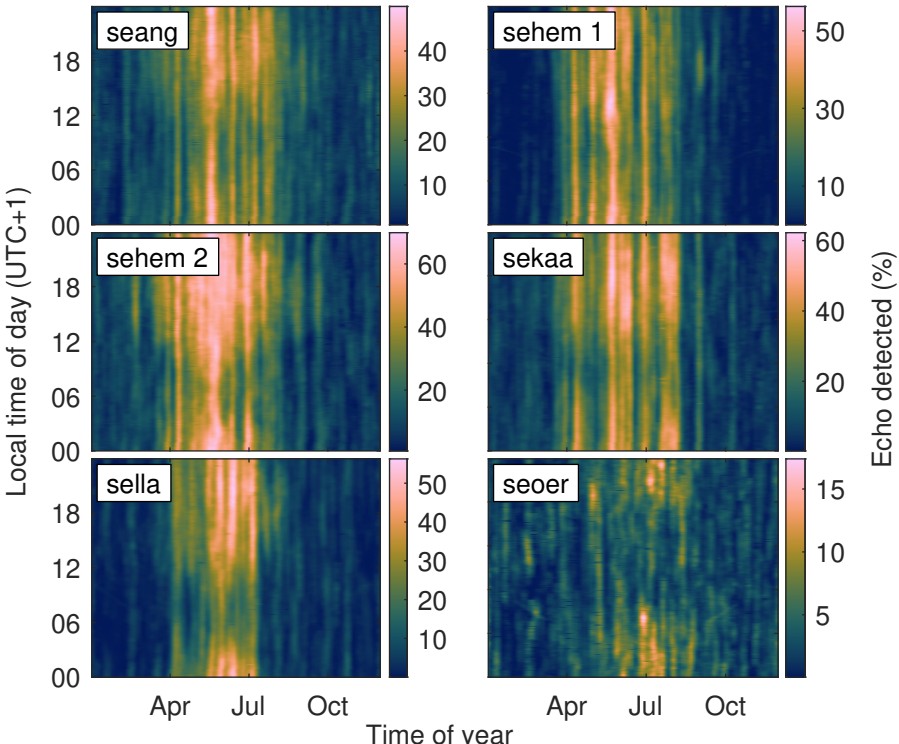

**Figure 9.** Relative frequency of detected echoes from all selected groups of radar cells as a function of the time of year and of the time of day.

coastal regions from the far side of nearby waters were selected. The temporal variation in the received ground clutter from these regions were analyzed.

The analyses show that a clear seasonal cycle exists in the extracted reflectivity data. During summer months (June, July, and August) ground clutter were more frequent and also stronger whereas during winter (December, January, and February) ground clutter were less frequent and much weaker.

In addition to the seasonal cycle a diurnal cycle was found. The diurnal cycle was much weaker than the seasonal cycle but it was nevertheless found that ground clutter were more frequent (and stronger) in the evening (around 18 PM, local time) and
less prevalent (and also weaker) in the morning (around 6–7 AM, local time).

Since the weather radars utilize dual polarization the data set was examined for any systematic difference in reflectivity data from horizontal and vertical polarizations. In the examined data set, reflectivity data from horizontal polarization were slightly stronger (up to ca 1 dB) than reflectivity data from vertical polarization.

*Competing interests.*   The author declares that he has no competing interests.





**Figure 10.** Columns 1–3 show histograms of pairwise measurements of reflectivity data using horizontal and vertical polarization for three different data sets: total reflectivity, Doppler filtered reflectivity, and the difference between the total and filtered data sets. Column 4 shows empirical distribution functions of the pairwise difference between horizontal and vertical measurements.



*Acknowledgements.* The radar data were collected by the Swedish Meteorological and Hydrological Institute (SMHI). The scientific color maps used in this work were created by Crameri (2021) and Brewer (2022). Figure 1 was generated using the mapping package M_Map (Pawlowicz, 2021). This work was financed by the The Swedish Defence Materiel Administration (FMV).



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
