# Peer review of "Observations of anomalous propagation over waters near Sweden"

_Atmospheric Measurement Techniques, 2022_

## Author Response (AR1)

**Author's response**

Here follows the point-by-point response to the both reviews, including relevant changes to the manuscript.

Please note that I have also revised a number of figures, according to requests from the reviewers. I upload the new figures as a supplement.

**Response to reviewer 1**

I thank the referee for the review of the manuscript. Here follows a point-by-point reply to the comments (reproduced in italics).

*Specific comments*

1. *P2 line 34, suggest: on radar systems -> on radar systems observations [or similar]*

   Done.

2. *P2 line 48. Here (or later in the paragraph) the text could cite Bech et al (2007) – see reference below – as they show observations and NWP simulations of a case study of intense superrefraction from the Hemse radar, similarly as in this manuscript.*

   The reference has been added to the manuscript together with the following text.

   "Bech et al. (2007) investigated the effects of anomalous propagation on beam blockage models and in a case study applied their method to data from the same weather radar on Gotland."

3. *P3 line 65. Did the author check if the -32 dBZ signal value can be observed at all ranges considered (up to 240 km), i.e. are the radars sensitive enough for that? In case this is not the case please comment possible effects on the results. Please clarify in the text.*

   The referee is correct that the radars cannot measure -32 dBZ at all ranges. The results are not affected by this other than that smaller effects of anomalous propagation can not be detected at large distances. The following text has been added to the manuscript:

   "At larger distances from the radar the sensitivity is not enough to measure -32 dBZ. For example, at the maximum distance of 240 km the minimum echo strength detected by the Swedish radars is approximately 10 dBZ."

4. *P4 line 69. Please clarify if the analysis is performed on polar volume data (radar pixels of 250 m x 1 degree) or in cartesian data.*

The stated range resolution is actually wrong, the correct range resolution is 500 m. This has now been corrected. All analyses were performed using polar volume data but only data from the scan with the lowest elevation angle (0.5°) were used. The following text has been added to the manuscript:

"All data in this work come from polar volumes but only data from the scan with the lowest elevation angle, 0.5°, were analyzed."

5. *P5 line 109. Please specify if the radar horizon shown in Fig. 2d is calculated considering the lower part of the 3 dB beamwidth or otherwise (for example the center of the beam). In any case, side lobes could still contribute to clutter, a possibility the text could mention.*

The radar horizon was calculated for the center of the beam. The following text has been added to the manuscript.

"The radar horizon, calculated for the center of the radar main lobe using standard atmospheric conditions, is also shown for comparison."

6. *P7 line 134. Can you please clarify here which time resolution are you using for the values stored as a function of time of day? It is not clear to this reviewer if it is the original 5 min resolution.*

The time resolution used was the original 5 min. This has now been clarified in the manuscript.

7. *P13 line 225. Suggest: environment -> atmospheric conditions [I think the term 'environment' is too general and the focus is really on the atmospheric conditions]*

Done.

8. *P14 line 231. Readers not familiar with the geographical region of study may not know where the Gulf of Bothnia is – this and other locations in the text such as Gotland could be indicated for example in Figure 1.*

Done.

9. *P14 line 238. The effect of subrefraction shouldn't be a decrease of detected ground clutter echo compared to normal propagation conditions, at least close to the radars?*

For large distances, as examined in this manuscript, no ground clutter is expected during standard atmospheric conditions. During subrefraction, I would also expect no radar echoes and hence it would be difficult to study anomalous propagation using the presented method. The following text has been added to the manuscript:

"Another limitation using this method is that it is not possible to account for the presence of subrefraction, at least not for the long distances studied here."

10. *P15 lines 250 – 255. The text could also comment on the possible relation of the diurnal and seasonal cycle of ground clutter with that of precipitation, in particular warm season convective precipitation.*

By subtracting the Doppler filtered radar data from the unfiltered radar data, echoes originating from precipitation are greatly suppressed (cf. Fig. 2). The impact of precipitation to the observed diurnal and seasonal cycles would therefore mainly consist of attenuation. Strong attenuation from heavy precipitation is rare in the studied areas and should not impact the observed cycles much.

*Technical Comments*

11. *P1, Abstract, suggest: can receive ground clutter -> can observe ground clutter [similarly in P2 line 42]*

Done.

12. *P1 line 10, Suggest: can also -> are also*

Done.

13. *P2 line 53. Please check citation style: should it be Essen et al., 2004, 2012 similarly as Danklmayer., … in the same line?*

The manuscript in prepared using LaTeX and references are therefore automatically typeset. I assume it is typeset this way because the author lists for Essen et al. 2004 and Essen et al. 2012 are different.

14. *P3 Table 2. Suggest: Gain -> Antenna gain*

Done.

15. *P3 line 71 and 72. I suggest converting the footnote information into references, quoted in the text and listed in the references section.*

Done.

16. *P6 Fig 2 caption. Island Gotland or Gotland island? Please check.*

Done.

17. *P7 line 132. Suggest: values = - 32 dBZ -> value = - 32 dBZ*

Done.

18. *P11 line 206 typo : In can be -> It can be*

Done.

**Response to reviewer 2**

I thank the referee for the review of the manuscript. Here follows a point-by-point reply to the comments (reproduced in italics).

1. *Line 33. Please define 'radar horizon' and specify how it was computed.*

   The following text and reference has been added to the manuscript.

   "… (for a description of the normal radio horizon, see, e.g., Skolnik (2001, Chap. 8))."

2. *Page 4. According to the AMT manuscript preparation guidelines, footnotes should be avoided in the text. I suggest using the references section instead.*

   Done.

3. *Page 6. Figure 2 shows averaged radar data, and for this reviewer, the sentence "total reflectivity data" may lead to misinterpretation, e.g. accumulated data. Consider rephrasing to 'unfiltered reflectivity data', 'raw reflectivity data' or similar.*

   Done.

4. *Line 156. Regarding the spectral analysis. Please specify how the data was transformed from the time domain to the frequency domain.*

   The following text has been added to the manuscript.

   "…, using the fast Fourier transform, …"

5. *Figures 3 and 4 show very interesting information. However, it isn't easy to analyse them in detail as the x-axis is too narrow. It would be better to scale these images to the page width.*

   Done.

6. *Line 218. Please replace 'ca' with $\sim$ or $\approx$.*

   Done.